# USING GENERATIVE AI TO CAPTURE HIGH FIDELITY TEMPORAL DYNAMICS TO TARGET VEHICULAR SYSTEMS

## ABSTRACT

Generative models have transformed the creation of text, images, and video content by enabling machines to generate high-quality, realistic outputs. These models are now widely being adopted in advanced fields like natural language processing, computer vision, and media production. Since vehicle data is limited due to proprietary concerns, utilizing generative models to mimic complex vehicle behaviors would provide powerful tools for creating synthetic data that can serve as a crucial component for enhancing the fidelity of vehicle models, better predictive maintenance, more robust control systems, autonomous driving features and resilient defense mechanism against cyber threats. This paper presents a Long Short-Term Memory (LSTM) based Conditional Generative Adversarial Network (GAN) model, which trains on limited available real vehicle data and is then able to generate synthetic time series data mimicking the actual vehicle data. The LSTM network helps in learning temporal characteristics of vehicle network traffic without needing the system details, which makes it applicable to wide range of vehicle networks. The conditional layer adds auxiliary information by labeling data for different driving scenarios for training and generating data. The quality of the synthetic data is evaluated visually and quantitatively using metrics such as Maximum Mean Discrepancy (MMD), Predictive and Discriminative Scores. For demonstration purposes, the generative model is integrated into a validated vehicle model, where it successfully generates synthetic sensor feedback corresponding to the dynamic driving scenarios. This showcases the model's ability to simulate realistic sensor data in response to varying vehicle operations. Leveraging the high similarity to actual data, the generative model is further demonstrated for its potential use as malicious attack mechanism due to its deception capabilities against state of the art Intrusion Detection System (IDS). Without triggering the thresholds of the IDS, the model is able to penetrate the network stealthily with a low detection rate of 47.05%, compared to the 90% or higher detection rates of other known attacks. This effort is intended to serve as a test benchmark to develop more robust ML/AI based defense mechanisms.

## 1 INTRODUCTION

Vehicle technology is evolving unprecedentedly, reshaping the transportation landscape by offering the promise of safer, more efficient, and sustainable mobility. Autonomous vehicles are now at the forefront of this transformation, powered by advanced sensors and control algorithms equipped with Machine Learning (ML) and Artificial Intelligence (AI) techniques, Howar and Hungar (2024). These algorithms are continuously trained and refined using vast amounts of data to improve decision-making, perception, and navigation capabilities. However, to ensure robust performance in a variety of real-world scenarios, from unpredictable traffic patterns to extreme weather conditions, these systems require diverse and comprehensive datasets.

Traditional data collection methods present several critical limitations when developing robust autonomous vehicle systems. Collecting and annotating real-world vehicle data is not only expensive and time-consuming but also constrained by the types of driving scenarios that can be encountered, Moveworks (2024). This limits the diversity of conditions in which these systems can be trained,

leaving gaps in preparedness for edge cases like extreme weather or rare traffic events. Moreover, real-world datasets often contain sensitive information, creating privacy and regulatory concerns that can further hinder data accessibility and sharing. Biases present in real-world data also challenge the generalization of machine learning algorithms, particularly when these systems must perform reliably in unpredictable or unfamiliar environments. To address these issues, synthetic data generation techniques are increasingly being employed Nikolenko (2021). By using these techniques, high-quality, diverse, and scalable datasets can be produced assisting in training the autonomous vehicle systems in a broader range of conditions, allowing them to handle rare or challenging driving situations with greater accuracy and reliability.

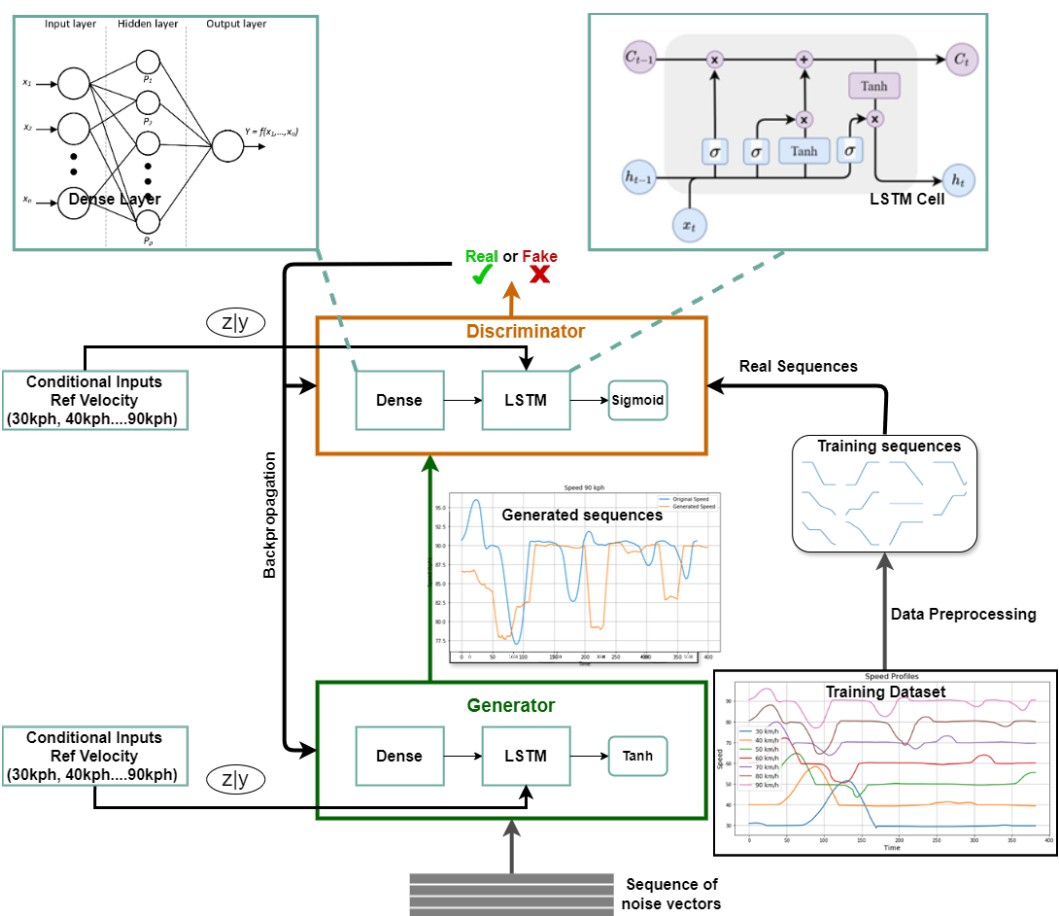

Figure 1: **LSTM-Conditional GAN Model**. The model takes driving profile sensor data (Training Data) for different Set-Point Commands (Conditional Inputs) as input. The Discriminator compares the real input data with the generated data and back-propagates its outcome. The Generator improves the generated data to match real data, based on the feedback from Discriminator, until it starts matching the real data.

In this paper, we propose a generative model based on LSTM based Conditional GAN, as shown in figure 1, for generating synthetic vehicle system behavior for a diverse range of driving profiles. Our focus is synthesizing the critical sensor information for different scenarios based on speed set-point commands using the generative model. Since the sensors exhibit physics following behavior correlating to the vehicle operation, each sample has some dependence on the previous sample hence making it a sequence following time series data. Recurrent Neural Networks (RNN) are the obvious choice to retain previous information and support in modeling this behavior intrinsically. However RNNs have limited retention capabilities which degrades with the increase in length of the data. LSTM, Hochreiter and Schmidhuber (1996), a special type of RNN, is selected to avoid the long-

term dependency problems. Another important factor to consider is the variation in vehicle operating modes, which can be categorized into distinct operational states. Each category represents a unique set of sensor behaviors and data corresponding to the vehicle's dynamic conditions. A basic GAN model, Goodfellow et al. (2014), however lacks the capability to generate data that reflects these categorical distinctions. To address this, incorporating conditional information into the model allows it to assign a label, $y$, to each driving cycle. This enables the GAN to generate synthetic sensor data that is specific to each operational category, improving the accuracy and relevance of the generated data for different driving scenarios.

The performance of the generative model is evaluated using three different metrics: Maximum Mean Discrepancy (MMD), Discriminative Score (DS) and Predictive Score (PS). Once trained, the generative model is integrated into a real-world validated vehicle model, Eriksson et al. (2016), to assess its performance and evaluate its potential use in future vehicle designs and models. The results showed that the model was able to accurately follow the vehicle's operational dynamics and generate synthetic sensor data that could be effectively used as input for the control algorithms, validating its applicability in real-world scenarios.

The effectiveness of the model is further evaluated by testing it against a state-of-the-art Intrusion Detection System (IDS), Kukkala et al. (2020). The IDS is first trained on real vehicle data and then tested using the generated synthetic data. It employs an auto-encoder, which detects discrepancies by calculating the reconstruction error. If the reconstruction error for the test input exceeds a pre-set threshold, established from the real data, the IDS triggers an alert, indicating potential discrepancy in the test input. The experimental evaluations show a detection rate of only 47% for the synthetic data compared to other types of injected data, which are detected at 100%. This result highlights a significant challenge in automotive cybersecurity, revealing that generative models could potentially be exploited to stealthily infiltrate and compromise even the most sophisticated systems and networks, posing threats to the safety and integrity of modern vehicles. Consequently, this research serves as a catalyst for developing more robust defense mechanisms to effectively counteract the persistent threat posed by the widespread integration of AI technology.

Overall, the main contributions can be summarized as:

- We develop a generative model specifically designed to learn the time series dynamics of a vehicle and is able to produce synthetic sensor data. The model is trained with conditional information using speed setpoint commands for different driving scenarios and is able to generate data on demand for these scenarios.

- We evaluate the quality of the generated data using three benchmark metrics: Maximum Mean Discrepancy (MMD), Discriminative Score (DS) and Predictive Score (PS), and receive satisfactory results

- We demonstrate the application of the generative model in vehicle operations by integrating it into a benchmark vehicle model. The generative model successfully follows the operational dynamics of the vehicle, showing its capability to generate realistic data that aligns with real-world vehicle behavior.

- We further demonstrate how these generative models could be exploited for malicious injection attacks, targeting the security of vehicle networks. Due to their low detection rate when tested against a state-of-the-art Intrusion Detection System (IDS), these models pose a significant threat, highlighting potential vulnerabilities in current automotive cybersecurity measures.

## 2 DESIGNING THE GENERATIVE MODEL FOR VEHICLE SYSTEMS

Vehicle systems can be defined as dynamic models that receive set-point commands either from the driver or, in the case of autonomous systems, from a supervisory control system. These systems generate control actions that drive the actuators to achieve the desired set-point based on feedback from sensors, ensuring the vehicle operates according to the intended commands and conditions. Using this information, we propose a generative model specifically designed for time-series data, corresponding to the temporal dynamics of vehicle sensors for different driving scenarios. Since the driving scenarios are defined by the set-point commands, we define each set-point command as the

conditional label $y_i(k)$, where $k \in \mathbb{Z}_+ := \{0, 1, \dots\}$, $i \in \{1, 2, \dots\}$ corresponds to the different speed set-point commands.

**Data Pre-Processing:** Since sensors produce complex physiological signals, accurately modeling them requires preserving the integrity of their temporal dynamics. To achieve this, the input training data must reflect the smoothness and continuity inherent in these signals, ensuring that the model captures their real-world, physics following behavior, effectively. We leverage the properties of Gaussian processes with a radial basis function (RBF) kernel, cmu . This kernel enforces local correlations between nearby points, reflecting the natural continuity observed in real sensor data. In our approach, we sample 30 equally-spaced points from the training data, representing sensor readings over time. This can be interpreted as drawing from a multivariate normal distribution, where the covariance between sensor readings is defined by an RBF kernel. By evaluating the covariance function on a grid of evenly spaced time points, we can specify the probability distribution underlying the real data. Each sensor type, denoted as $x_j(k)$, is included in the dataset as a discrete-time real sample, where $j \in \{1, 2, \dots\}$ represents different sensor types, and $x_j(k) \in \mathbb{R}^n$ denotes the sensor readings. The variable $z$ refers to the sequence of unstructured noise vectors in latent space. Overall, the training dataset is organized in an $X_{i \times j}$ matrix, where $i$ represents all the driving scenarios $y_i(k)$ based on the set-point command and $j$ represents the measurement vectors for all sensors $x_j(k)$:

$$
X_{i \times j} = \begin{bmatrix} x_{11} & x_{12} & \dots & x_{1j} \\ x_{21} & x_{22} & \dots & x_{2j} \\ \vdots & \vdots & \ddots & \vdots \\ x_{i1} & x_{i2} & \dots & x_{ij} \end{bmatrix}
$$

**LSTM networks to learn Temporal Dynamics:** Once the smoothness and local correlations in the training data are ensured, the LSTM network within the GAN can effectively capture the underlying features of the data. The LSTM cell is designed to retain and predict long-term dependencies, making it well-suited for time-series data. In the generator, a stacked LSTM architecture with 100 hidden units per layer is employed to generate physiological signals. Prior to the LSTM layer, a 2D categorical embedding layer and a linear layer are used to learn the labels of the set-point commands, $y$ during adversarial training. The mapping from the random latent space is accomplished through a dense layer with a tanh activation function, followed by the LSTM layer. In the discriminator, the label information is initially passed through the same 2D embedding layer and then upsampled via a dense layer before being concatenated with the input sequences. Both the generator and the discriminator use a repeat vector layer to expand the temporal dimensions, ensuring that the output matches the required number of time samples.

**Designing Generator and Discriminator Models:** The generator function, $G(z, y)$, generates realistic samples by taking noise $z$ and the conditional label $y$ as inputs. The discriminator, denoted by $D$, operates through two key functions: $D(x, y)$, which evaluates real data $x$ conditioned on label $y$, and $D(G(z, y), y)$, which assesses the fake data generated by the generator $G(z, y)$. The discriminator's role is to distinguish between real data from the dataset and synthetic data produced by the generator. The generator's objective, on the other hand, is to deceive the discriminator by producing data that becomes indistinguishable from real data. During training, the discriminator provides feedback to the generator through backpropagation, updating the generator's parameters based on the derivatives of the discriminator's output. This iterative process continues as the two models compete, with the ultimate goal of reaching a Nash equilibrium, where the discriminator can no longer differentiate between real and generated data.

The two adversarial models, generator and discriminator, engage in a min-max game, where the generator learns the data distribution, and the discriminator evaluates the authenticity of the generated samples. The discriminator's primary objective is to maximize the loss function $L_D$ to make $D(G(z, y), y)$ close to 0 and $D(x, y)$ close to 1:

$$
\max_D L(D) = \mathbb{E}_{x \sim p_{\text{data}}(x|y)}[\log D(x, y)]
$$
$$
+ \mathbb{E}_{z \sim p_z(z), y \sim p_y(y)}[\log(1 - D(G(z, y), y))] \tag{1}
$$

Conversely, the generator's objective is to mimic the underlying features of real data and produce convincing fake samples by minimizing the loss function $L_G$ to make $D(G(z, y), y)$ close to 1:

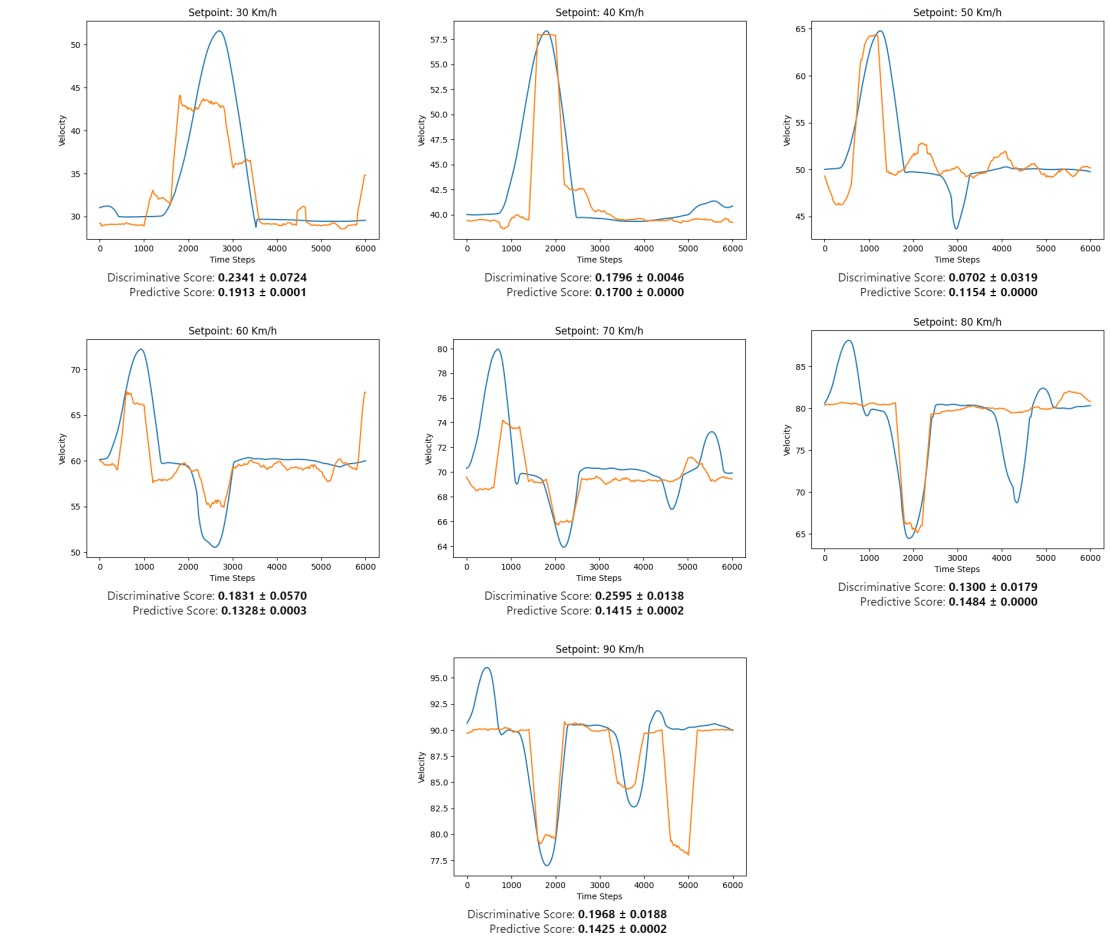

Figure 2: Comparison of original (blue line) and generated (orange line) data for each velocity setpoint, with corresponding discriminative and predictive scores.

$$\min_G L(G) = \mathbb{E}_{z \sim p_z(z), y \sim p_y(y)}[\log(D(G(z, y), y))] \qquad (2)$$

## 3 EVALUATION OF THE GENERATIVE MODEL

To evaluate our generative model, we have established multiple criteria that encompass both qualitative and quantitative approaches. These criteria include performance metrics that assess the model's accuracy and reliability, as well as application-based effectiveness that evaluates its practical utility in real-world scenarios, i.e. 1. Deceiving an Intrusion Detection System, 2. Operating in a Vehicle Model.

### 3.1 EVALUATING USING PERFORMANCE METRICS

We first assess the fidelity and quality of the synthetic time-series data generated by the model according to three different metrics commonly used in the literature: Maximum Mean Discrepancy (MMD), Discriminative Score (DS) and Predictive Score (PS).

*Maximum Mean Discrepancy:* MMD Gretton et al. (2012) quantifies the similarity of two distributions $p(x)$ and $q(y)$ by evaluating the distance between their Hilbert space mean embeddings. Such a measure can be empirically estimated from a finite number of samples. Given $\{x_i\}_{i=1}^N \sim p(x)$

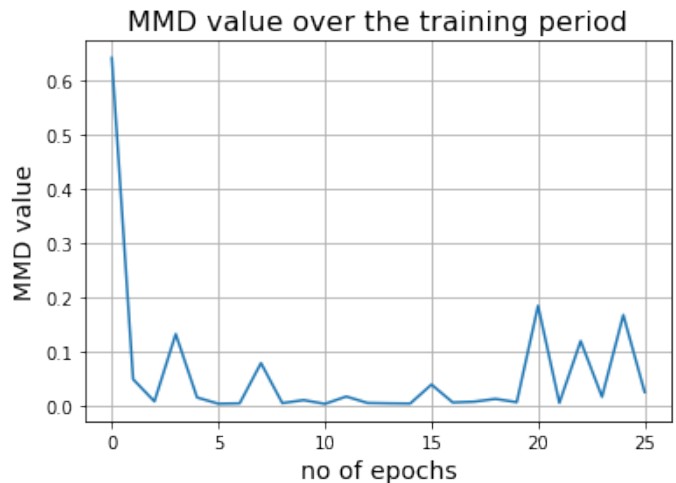

Figure 3: Maximum Mean Discrepancy during training.

and $\{y_j\}_{j=1}^M \sim q(y)$, an estimate of MMD is:

$$MMD = \left\{ \frac{1}{N^2} \sum_{i=1}^N \sum_{j=1}^N K(x_i, x_j) - \frac{2}{MN} \sum_{i=1}^N \sum_{j=1}^M K(x_i, y_j) + \frac{1}{M^2} \sum_{i=1}^M \sum_{j=1}^M K(y_i, y_j) \right\}^{1/2} \quad (3)$$

where $K(x,y) = exp(-\|x - y\|^2/2\sigma^2)$ is the Radial Basis Function (RBF) kernel. After each training epoch, we generate a thousand samples and compute the MMD against the held out test data. The resulting curve is shown in figure 3. The MMD gradually decreases and converges relatively quickly as training goes on. This indicates that the probability distribution of the synthetic data generated by the model approaches and gets very close to the real data distribution.

*Predictive and Discriminative Score:* these two metrics were first introduced by Yoon, Jarrett and Van der Schaar Yoon et al. (2019) as a mean to quantify the fidelity, diversity and usefulness of synthetic time series data produced by generative models.
The DS is the classification error of a post-hoc 2-layer LSTM model trained to distinguish between real and synthetically generated time sequences. First, real sequences are labeled *real* and synthetically generated sequences are labeled *fake*, then the model is trained. Finally, the DS is computed as follows:

$$DiscriminativeScore = |0.5 - Acc| \quad (4)$$

where $Acc$ is the classification accuracy of the model on a held-out test set.
The PS is derived through the optimization of a 2-layer LSTM model, which predicts the value of the upcoming time step for each input sequence. This model is trained using synthetically generated data and subsequently tested on real data, with its performance assessed in terms of Mean Absolute Error (MAE).

Figure 2 displays a comparison of original and generated time series for each of the 7 velocity set-points, along with the respective Discriminative and Predictive scores obtained in our experiments. For all the velocity profiles, the values remain consistent and comparable to those reported by Yoon, Jarrett and Van der Schaar in their original paper Yoon et al. (2019). This is another indication of the model being able to successfully learn the distribution of the original velocity dataset.

## 3.2   EVALUATION USING AN INTRUSION DETECTION SYSTEM

We establish a validation criterion for our proposed generative model by testing whether it can consistently bypass detection by a state-of-the-art AI-based Intrusion Detection System (IDS), which is specifically trained to identify anomalies or discrepancies in normal data. We shortlisted a Recurrent Autoencoder-based IDS built on Gated Recurrent Units (GRUs), named INDRA Kukkala et al. (2020), due to its superior performance in detecting anomaly attacks on critical cyber physical

systems. More precisely, during the training process, the Autoencoder learns and tunes its weights on the temporal relationships that exist between the series of signal values characterizing normal behavior. This allows it to reconstruct normal data with high fidelity. At the same time, the model will struggle to reconstruct data which significantly deviates from normal traffic. This property is used to detect anomalies at run time by monitoring an Intrusion Score (IS), defined as the square of the stepwise reconstruction error. When the error exceeds a certain threshold the data is classified as anomalous. In this case, the threshold was set as the highest stepwise reconstruction error registered on the test set.

In practice and to avoid complications, we trained the generative model using data from the vehicle's velocity sensor, corresponding to various velocity set-point commands. The training set consisted of speed profiles based on 7 distinct velocity set-points, ranging from 30 km/h to 90 km/h in 10 km/h increments, with approximately 40,000 samples over 380 seconds of vehicle operation. 6 of them were used for training and the last one was used as test set. Furthermore, to benchmark the performance of the IDS and provide a meaningful comparison for our generative model, we utilized standard anomalies and attacks commonly referenced in the literature for vehicle networks, including sawtooth, random, plateau, and replay attacks. More details related to the evaluation can be found in OSU-Cyberlab (2024).

Table 1: INDRA IDS Detection Accuracy

| Attack Types | Detection Accuracy |
|---|---|
| Random Attack | 100% |
| Sawtooth Attack | 91.18% |
| Plateau Attack | 88.24% |
| Replay Attack | 91.18% |
| **Proposed Generative Model** | **47.05%** |

Notably, the IDS achieves very high detection accuracy on sawtooth and random attacks 91.18% and 100% respectively, and above 88.24% & 91.18% detection accuracy on plateau and replay attacks. However, it struggles to identify the data from generative model, detecting only 47.05% of the cases. Hence making it more stealthier if to be used as a potential cyberattack for malicious data injection. For comparison purpose and to prevent over complicating the figure, plateau attack was used (since it had the lowest detection rate among all the other known attacks) against the generative model. Figure 4 demonstrates the evaluation process, where figure 4 (a) shows the cases of Plateaus and Generative model data given as input to the IDS and the corresponding reconstruction of the signal by the IDS, and figure 4 (b) shows the corresponding Intrusion Score (IS) evaluated based on the reconstruction error. The red highlighted background indicates the region where the malicious data injected and evaluated by the IDS. When the plateau attack is introduced, the reconstructed signal deviates significantly from the actual one, causing the IS to cross the threshold. In contrast, with the generative model data, the reconstruction error stays within the threshold, and the IS plot remains nearly flat, avoiding the triggering of any alarm.

### 3.3 APPLICATION OF GENERATIVE MODEL IN A VEHICLE

One of the key applications of the proposed generative model lies in its ability to synthesize sensor data for dynamic vehicle operations. By generating realistic and high-fidelity sensor outputs, the model can be leveraged to train and evaluate advanced architectures in automotive systems, particularly for applications involving autonomous driving, network security, and control optimization.

The model's performance is demonstrated under a dynamic driving scenario, as illustrated in figure 5. In this test, the vehicle begins from a stationary position and accelerates to a steady-state velocity of 90 km/h based on the set-point command (solid blue line). At $t = 150$ seconds, the generative model starts producing synthetic sensor data (solid red line) to mirror real-time sensor feedback (magenta dashed line). To further assess the model's robustness and adaptability, the set-point command is periodically reduced by 10 km/h at 100-second intervals. As the vehicle transitions through these varying speed profiles, the model continuously tracks and adjusts to the changes, generating accurate sensor outputs in response to each new set-point command. This ability to adapt ensures that the synthetic sensor data aligns closely with real-world driving dynamics, making the model a

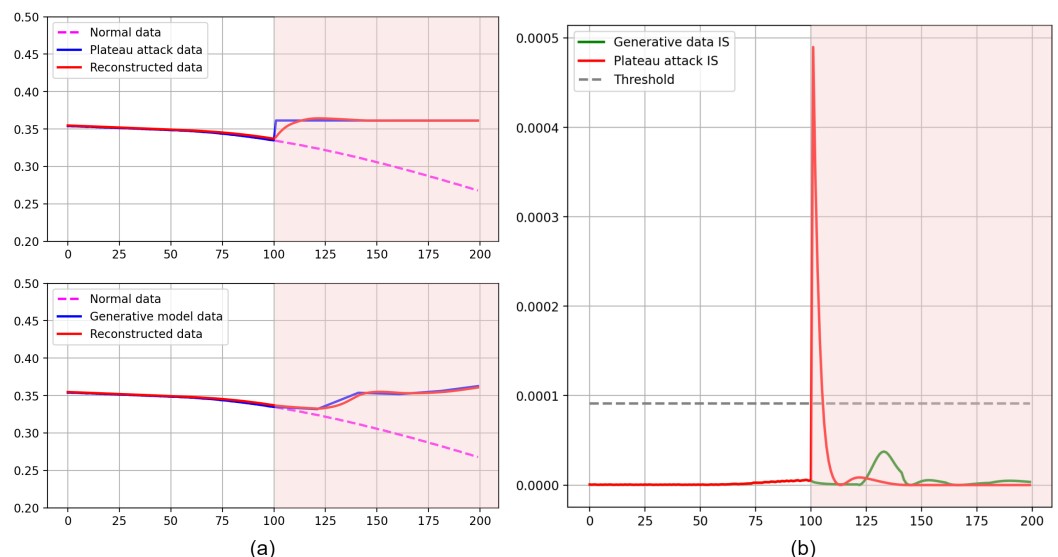

Figure 4: Snapshot of IDS checking a signal under 2 different attacks, i.e. Plateau and Generative Model input. (a) shows the signal comparison (where the normal profile is also shown for reference) and (b) the corresponding Intrusion Score.

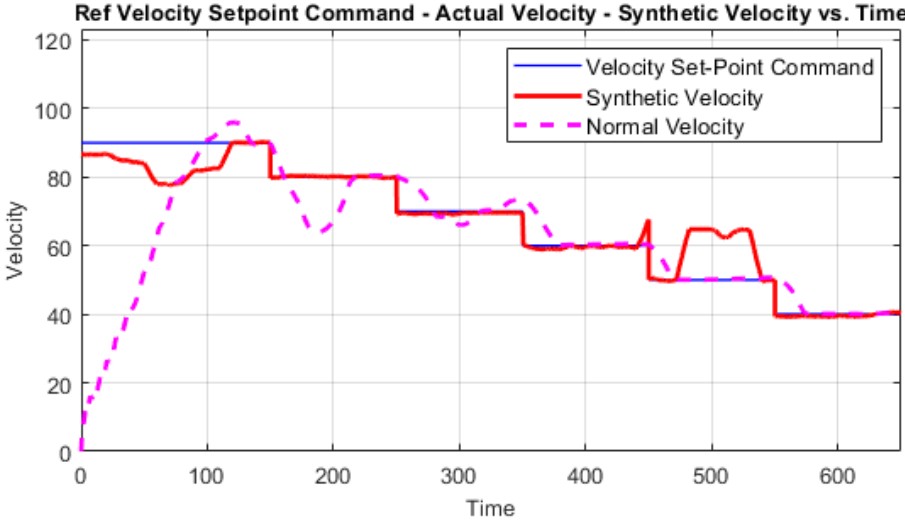

Figure 5: Performance evaluation of Generative Model producing synthetic velocity sensor value (red line) for the desired set-point command (blue line).

valuable tool for testing and refining vehicle control algorithms and network protocols in a variety of conditions.

## 4 RELATED WORK

**Generative AI in Vehicles.** Generative AI techniques have been gaining significant traction in the field of automotive cybersecurity. Recent advancements have led to the development of novel Intrusion Detection Systems (IDSs) using Generative Adversarial Networks (GANs). For example, Seo et al. (2018), Chen et al. (2021), and Kavousi-Fard et al. (2020) introduced GAN-based IDSs capable of detecting both known and unknown ID-based attacks, achieving detection accuracy rates as high as 100%. Additionally, Desta et al. (2020) proposed an LSTM-based IDS that identifies

anomalies in Network ID sequences by comparing predicted IDs with actual ones. Similarly, the works of Tanksale (2020) and Hanselmann et al. (2020) utilized LSTM-based models to predict the next valid network sample and detect anomalies by analyzing deviations from the predicted values.

However, attackers have also started leveraging Generative AI to their advantage. They use these techniques to craft malicious payloads, generate harmful code snippets, and even compile them into executable malware files. As highlighted by cha (2023a) and cha (2023b), this dual-use of generative AI poses new challenges, as it enables the creation of sophisticated cyberattacks that can evade traditional detection mechanisms.

**GANs to generate Time-Series Data.** Esteban et al. (2017) proposed Recurrent GAN model specifically to generate medical data, Smith and Smith (2020) proposed Time Series GAN (TSGAN) using "few shot approach". Ehrhart et al. (2022) proposed a Convolution Network based GAN for their application of wearable sensors. Saravana et al. (2024) proposed a Bi-LSTM architecture for GANs specifically designed to address forced oscillation (FO) source localization in power systems. These works have been our primary source of inspiration to design generative model to synthesize vehicle sensor data.

## 5 CONCLUSIONS

In this paper, we proposed an LSTM based GAN model to generate sequential time-series data that mimics the temporal dynamics of actual vehicle sensor data. We have demonstrated the feasibility of using this generative model to simulate dynamic vehicle operations by learning the temporal relationships between sensor data and control commands. Our model can produce highly realistic synthetic sensor data, which can be used to train and evaluate advanced vehicle systems and security frameworks. The effectiveness of the model has also been demonstrated as potential stealthy attack mechanism against a state-of-the-art IDS, which sets the stage to use it as test-bed to develop more resilient defence mechanisms. Some limitations of the proposed model are highlighted here for future work:

- **Limited Generalization Across Diverse Scenarios:** The generative model is trained on a specific set of driving conditions (e.g., a limited range of velocity set-point commands). This may limit its ability to generalize to unseen or more complex driving scenarios (e.g., aggressive maneuvers, extreme weather conditions, or unusual traffic patterns). Future work could explore training the model on a broader dataset to enhance its versatility.

- **Sensitivity to Training Data Quality:** The quality of the synthetic data is highly dependent on the quality and variety of the real data used for training. If the training data does not fully represent the operational scenarios of a vehicle, the generative model might produce inaccurate or incomplete synthetic data. More comprehensive datasets or data augmentation techniques could mitigate this issue.

- **Scalability and Computational Complexity:** As vehicle systems become more complex, the generative model might face challenges in scaling efficiently. Training and maintaining high performance across multiple sensors and vehicle subsystems (e.g., LiDAR, radar, cameras) would require more computational resources, possibly hindering the model's scalability.

- **Ethical and Security Implications:** Although the generative model has valuable applications, its misuse as a cyberattack tool raises ethical and security concerns. Future research should focus on developing safeguards to ensure the technology is used responsibly and does not become a tool for malicious data injection or system disruption.

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

```
dvips mypaper_iclr.dvi -t letter -Ppdf -G0 -o mypaper_iclr.ps
ps2pdf mypaper_iclr.ps mypaper_iclr.pdf
```

