# OpenReview forum: "Using Generative AI to capture High Fidelity Temporal Dynamics to target Vehicular Systems"
_ICLR.cc/2025/Conference — ICLR 2025 Conference Withdrawn Submission_

### Official Review · Reviewer_TeHJ · 2024-10-27

**Soundness:** 3
**Presentation:** 2
**Contribution:** 2
**Rating:** 3
**Confidence:** 4

**Summary:**

The paper proposes a generative model based on a Long Short-Term Memory (LSTM) Conditional Generative Adversarial Network (GAN) to generate synthetic time series data for vehicle systems. It aims to create realistic sensor feedback data to simulate different driving scenarios.

**Strengths:**

The paper proposes to use generative AI models for vehicular systems. By focusing on the potential of generative models to simulate realistic vehicle behavior, the authors are tackling a key issue in improving vehicular simulations and evaluating security risks.

The paper uses a set of established metrics (Maximum Mean Discrepancy, Discriminative Score, and Predictive Score) to evaluate the quality of the generated data.

**Weaknesses:**

The contribution is limited. The use of LSTM networks and Conditional GANs to generate synthetic time series data is not a new concept. The paper's contribution is largely a re-application of existing ideas without substantial innovation. For instance, the LSTM architecture for capturing temporal dependencies in time series data has been well-established, and adding conditional inputs (like vehicle speed set points) is a straightforward extension rather than a novel breakthrough. The paper fails to explain why this combination of methods is unique or how it advances the field.

The evaluation needs more justification. Autonomous vehicles rely on multiple sensors that operate under complex and varied conditions. A more thorough evaluation would test the model under different driving scenarios, sensor types, and environmental conditions. For example, how would the model perform if trained on multi-sensor data from different operating conditions, like rain, fog, or night driving?

**Questions:**

Why choose LSTM instead of other RNN models?

How do we deal with the training stability of the generative models?

How to guarantee the completeness of the conditional information for the data generation?

---

### Official Review · Reviewer_aHbA · 2024-11-01

**Soundness:** 2
**Presentation:** 2
**Contribution:** 2
**Rating:** 1
**Confidence:** 5

**Summary:**

This paper proposes a generative framework to synthesize the vehicle system data, e.g., sensors data in extreme weather or conditions, addressing data scarcity issues in vehicle modeling. By learning temporal dynamics and incorporating labels for different driving scenarios, the model produces high-fidelity data that mimics real vehicle behavior, useful for autonomous driving, maintenance, and cybersecurity. The authors evaluate their approach with fidelity metrics and tested against an Intrusion Detection System (IDS). The model demonstrates its ability to closely resemble real data and evade IDS detection in 47%, indicating a potential security risk.

**Strengths:**

1. The LSTM-based Conditional GAN effectively generates high-fidelity synthetic data that closely mimics real vehicle sensor behavior, making it useful for a variety of applications.
2. Scenario-Specific Modeling: The usage of cGAN in different scenario is intuitive but effective. By incorporating conditional labels, the model can produce data for different driving scenarios, enhancing its relevance and applicability in training and testing vehicle systems under various conditions.
3. The model reveals vulnerabilities in Intrusion Detection Systems (IDS) by evading detection in 47% of cases, revealing the potential cybersecurity threats and the need for more robust IDS against generative AI-powered attacks.

**Weaknesses:**

1. There is limited technical contribution because all the network framework and losses are already proposed, I didn't see much novelty contribution of this work.
2. The experiment is setting is quite simple.
3. As stated in the conclusion, the proposed work does not resolve the generalization, training data quality, and real-world problems.

**Questions:**

Thanks for submitting this work to ICLR. I appreciate the authors input to bring cGAN to generate synthetic sensor data to facilitate autonomous driving, however, I have several concerns for this work:

1.  In the introduction, the author claimed that "there is a gaps in preparedness for edge cases like extreme weather or rare traffic events", I was expected to see technique contribution and experimental efforts to collect data and resolve the problem, however, I was disappointed to find there is no such presentation in this work.

2. The author overclaimed the contribution by stating generate multiple sensors data and even put a matrix to clarify the $X_{ij}$, however, I didn't find any contribution to enforce the generalization of generating multiple sensors data. In experiment, there is only one sensor data is synthesized, which is not aligned with the author's claim.

3. I am actually quite confused about the setpoint setting, in the real world, there is no such command to set a discrete velocity, in stead, there is only involve driver's action like press the pedal or break, which should be continuous action with continuous feedback. Therefore, I am convinced by the usage of the model in the autonomous driving scenario.

4. Fig. 5 is also confused to me, as the synthetic velocity always align with the set-point command and away from the normal velocity, how it is possible for a real car to have the speed like this (e.g., sudden speed drop)? And how this will contribute to the decision making of autonomous driving?

5. There is no source code or demonstration to support this finding.

---

### Official Review · Reviewer_Gfga · 2024-11-03

**Soundness:** 2
**Presentation:** 2
**Contribution:** 1
**Rating:** 3
**Confidence:** 3

**Summary:**

The paper proposes a Long Short-Term Memory (LSTM) based Conditional Generative Adversarial Network (GAN) model to generate sequential time-series data of the temporal dynamics of actual vehicle sensor data. Their model produces highly realistic synthetic sensor data, which can be used for training and evaluating advanced vehicle systems and security frameworks. Their evaluation demonstrates that their generated data can bypass state-of-the-art intrusion detection systems (IDS) and claim more resilient defense mechanisms are needed.

**Strengths:**

The paper designs the LSTM-based GAN model to generate time-series data that can bypass state-of-the-art IDS methods. I agree with their research motivation to improve cybersecurity in automotive systems with generative models. As of the recent huge progress in generative models, there should be a large potential to enhance security.

**Weaknesses:**

I have the following major concerns on this paper:

### Lack of discussion and evaluation of recent line of research in generative diffusion models

This paper should discuss and evaluate the recent generative models for automotive more clearly. This research is motivated by the limitation of existing works in high annotation cost of real-world data and incapability in covering edge cases like extreme weather or rare traffic events. These are the same motivations for the recent generative model works such as [a, b, c]. I can see that this work focuses on the sensor input generation rather than scenario generation, but scenario generation should be preferable as long as it is realistic since we still do not know what kind of driving scenario is required to reproduce the sensor input. To address these concerns, this paper should discuss the differences between their methods and these recent generative approaches and demonstrate their advantages quantitatively. At least, I want to know why this paper avoids even discussing such important prior work.

[a] Hu, A., Russell, L., Yeo, H., Murez, Z., Fedoseev, G., Kendall, A., ... & Corrado, G. (2023). Gaia-1: A generative world model for autonomous driving. arXiv preprint arXiv:2309.17080.
[b] Zhang, W., Wang, G., Sun, J., Yuan, Y., & Huang, G. (2024). STORM: Efficient stochastic transformer based world models for reinforcement learning. Advances in Neural Information Processing Systems, 36.
[c] Bogdoll, D., Yang, Y., & Marius Zöllner, J. (2023). Muvo: A multimodal generative world model for autonomous driving with geometric representations. arXiv e-prints, arXiv-2311.


### Lack of sufficient evaluation of vehicular system data

This paper should provide a more sufficient evaluation of vehicular system data because the current evaluation does not answer if these signals are realizable in actual driving scenarios. Exploring edge cases is important, but edge cases are meaningless if they are not realizable. The current evaluations with MMD, DS, and PS do not give a good understanding of vehicle-level consequences. I understand that it may not always be possible to come up with actual driving scenarios, but, this paper should provide at least several use cases. Otherwise, this work cannot provide meaningful feedback to the developers.


### Limited novelty in LSTM-based GAN model for generating time-series data

As this paper mentions, their methodology is highly inherited from Yoon et al. (2019), which originally proposed a time-series data generation with GAN and LSTM. Although the application to new areas can be seen as a contribution, it can only be seen as a minor contribution unless sufficient evaluation is provided to illustrate that the application brings significant improvements in the areas. If this paper intends to claim the novelty in their methodology, this paper should highlight their technical updates and their motivations for why these updates are necessary in this research.

**Questions:**

- Could you elaborate more about the position of this paper against recent generative diffusion models?
- Did you confirm the realizability of their attack in the real world? If so, how did you do that?
- What is the key design contribution of their method over Yoon et al. (2019)?

**Details Of Ethics Concerns:**

No concerns

---

### Official Review · Reviewer_fMLL · 2024-11-03

**Soundness:** 2
**Presentation:** 2
**Contribution:** 2
**Rating:** 3
**Confidence:** 3

**Summary:**

This work presents a Long Short-Term Memory (LSTM) based Conditional Generative Adversarial Network (GAN) model designed to generate synthetic time-series data that mimics real vehicle sensor data. Specifically, the model aims to learn temporal characteristics of vehicle network traffic without detailed system knowledge, making it applicable across various vehicle networks.

**Strengths:**

1. The paper addresses the challenge of limited access to vehicle data due to proprietary concerns. By generating high-fidelity synthetic data, it contributes to enhancing vehicle models, predictive maintenance, and the development of more robust control systems.

2. Leveraging LSTM networks allows the model to capture the temporal dependencies inherent in vehicle sensor data, which is critical for realistic time-series generation.

3. The exploration of the model's ability to evade IDS provides valuable insights into potential vulnerabilities in current security systems.

**Weaknesses:**

The application of LSTM-based Conditional GANs to time-series data is not novel. Prior works, such as [1, 2, 3], have proposed the use of LSTM-GANs for time-series prediction.

[1] Mogren, O. (2016). C-RNN-GAN: Continuous recurrent neural networks with adversarial training. arXiv preprint arXiv:1611.09904.

[2] Yu, Y., Srivastava, A., & Canales, S. (2021). Conditional LSTM-GAN for melody generation from lyrics. ACM Transactions on Multimedia Computing, Communications, and Applications (TOMM), 17(1), 1-20.

[3] Rao, J., Gao, S., Kang, Y., & Huang, Q. LSTM-TrajGAN: A Deep Learning Approach to Trajectory Privacy Protection}}. In 11th International Conference on Geographic Information Science (GIScience 2021) (Vol. 177, No. Part I, p. 12). Schloss Dagstuhl--Leibniz-Zentrum f {\" u} r Informatik.

**Questions:**

From my perspective, the performance in evading detection by state-of-the-art Intrusion Detection Systems (IDS) depends on the quality of the generative model, specifically the LSTM-GAN proposed in the paper. However, LSTM-GAN [1,2,3] is not a novel concept; it was introduced quite some time ago, and there are now more advanced models available, such as diffusion model for time series generation[4,5]. Therefore, the contribution of this paper appears seems to be incremental. Could the author clarify how this work contributes beyond the referenced studies?

[4] Lin, L., Li, Z., Li, R., Li, X., & Gao, J. (2024). Diffusion models for time-series applications: a survey. Frontiers of Information Technology & Electronic Engineering, 25(1), 19-41.

[5] Yang, Y., Jin, M., Wen, H., Zhang, C., Liang, Y., Ma, L., ... & Wen, Q. (2024). A survey on diffusion models for time series and spatio-temporal data. arXiv preprint arXiv:2404.18886.

---

### Note · Authors · 2024-11-27

I have read and agree with the venue's withdrawal policy on behalf of myself and my co-authors.